# Application of a SARS-CoV-2 Antigen Rapid Immunoassay Based on Active Microfluidic Technology in a Setting of Children and Young Adults

**DOI:** 10.3390/v16010041

**Published:** 2023-12-26

**Authors:** Christian Leli, Lidia Ferrara, Paolo Bottino, Cristina Bara, Iacopo Megna, Serena Penpa, Enrico Felici, Antonio Maconi, Andrea Rocchetti

**Affiliations:** 1Microbiology and Virology Laboratory, A.O. “SS. Antonio e Biagio e C. Arrigo”, Via Venezia 16, 15121 Alessandria, Italy; christian.leli@ospedale.al.it (C.L.); lidia.ferrara@ospedale.al.it (L.F.); cbara@ospedale.al.it (C.B.); arocchetti@ospedale.al.it (A.R.); 2Research and Innovation Department (DAIRI), A.O. “SS. Antonio e Biagio e C. Arrigo”, Via Venezia 16, 15121 Alessandria, Italy; iacopo.megna@ospedale.al.it (I.M.); serena.penpa@ospedale.al.it (S.P.); amaconi@ospedale.al.it (A.M.); 3Pediatric and Pediatric Emergency Unit, A.O. “SS. Antonio e Biagio e C. Arrigo”, Via Venezia 16, 15121 Alessandria, Italy; enrico.felici@ospedale.al.it

**Keywords:** COVID-19, SARS-CoV-2, microfluidic, POCT, pediatric emergency medicine

## Abstract

To carry out effective and quick identification of SARS-CoV-2 from nasopharyngeal swabs and contain outbreaks, reliable and rapid tools are needed. Herein, we compared a rapid antigen test based on active microfluidic technology to an RT-qPCR assay in pediatric and young adult patients admitted to the Pediatric Emergency Unit of a Children’s Hospital. Nasopharyngeal swabs collected from patients with suspected COVID-19 disease and from those without COVID-19 related symptoms, but requiring hospitalization, were performed with both antigen test and RT-qPCR assays. We included 375 patients with a median age of 5 years in the study, with an estimated overall prevalence of 7.2%. Overall, we observed a specificity of 97.4% (95% CI: 94.9–98.7) and a sensitivity of 66.6% (95% CI: 46.0–82.7) with a positive likelihood ratio (LR+) of 25.8 (95% CI: 12.8–51.8). In the subgroup of symptomatic patients, the specificity and the sensitivity were 95.2% (95% CI: 89.4–98.0) and 80.0% (95% CI: 44.2–96.5) respectively; LR+ was 16.6 (95% CI: 7.19–38.6). In the asymptomatic subset, the performance showed a specificity of 98.7% (95% CI: 95.8–99.7), a sensitivity of 58.8% (95% CI: 33.5–80.6), and an LR+ of 43.7 (95% CI: 13.3–144.0). Compared to RT-qPCR, the new microfluidic-based antigen test showed higher specificity (>95%) in the pediatric population, thus representing a suitable point-of-care testing (POCT) in a clinical setting with low prevalence of COVID-19.

## 1. Introduction

In December 2019, an outbreak of viral pneumonia caused by a new pathogen firstly classified as the novel coronavirus 2019 (2019-nCoV) was identified in Wuhan, China [1]. The 2019-nCoV was then reclassified as severe acute respiratory syndrome coronavirus 2 (SARS-CoV-2) [2] and the disease named coronavirus disease 2019 (COVID-19). In just a few months, COVID-19 became a global pandemic [3].

The gold standard method for the detection of SARS-CoV-2 has been quantitative reverse transcriptase polymerase chain reaction (RT-qPCR) testing [4]. However, it is expensive, time-consuming and required skilled technicians; in contrast, rapid antigen tests (RDTs) for direct detection of SARS-CoV-2 (whole virus, antigen, nucleic acid) or anti-SARS-CoV-2 antibodies were becoming more common in both hospital and community settings [5,6,7]. Their sensitivity and specificity compared to the RT-qPCR assay ranged from 37% to 90% and 65% to 100%, respectively, with a correlation between symptomatic/asymptomatic patients and cycle threshold (Ct) values [8]. RDTs were classified as first- and second-generation manual lateral flow assays and laboratory-based fourth-generation chemiluminescent immunoassays. Among them, the third-generation microfluidic assays (also known as lab on a chip) represented an ideal tool for simplifying laboratories processes on a tiny device able to integrate sample handling, analysis, and other processes with benefits such as rapid detection and low cost [9,10]. A microfluidic assay normally consists of a chip, an analyzer, a drive source, and a signal detection device that can miniaturize the several steps involved in sample detection, requiring a much smaller volume of biological specimens [11]. Due to their several advantages, they have played a significant role in increasing infection detection rates and facilitating better healthcare services.

However, fewer data about the diagnostic performance of RDTs in pediatric populations were available [12,13,14], since adults represented the main affected age group with higher morbidity and mortality. Furthermore, it emerged that children played a minor role in the dynamics of the COVID-19 pandemic [15].

In order to carry out effective and rapid identification of SARS-CoV-2-positive patients among pediatric and young adult subjects, reliable and rapid testing tools are needed [16].

The aim of this study was to compare the diagnostic accuracy of a new RDT based on microfluidic technology to conventional RT-qPCR testing in a low-prevalence setting consisting of pediatric/young adults patients admitted to the pediatric emergency unit of a tertiary pediatric hospital in Italy.

## 2. Materials and Methods

### 2.1. Design of the Study

From January 2021 to November 2021, 375 patients aged <18 years admitted to the pediatric emergency unit of the Cesare Arrigo Children’s Hospital in Alessandria (Italy) were enrolled in the present study and tested with nasopharyngeal swabs. The following data were collected for each patient: (I) medical history and physical examination, focusing on signs or symptoms related to COVID-19 [17]; (II) results of nasopharyngeal swabs for SARS-CoV-2 RNA detection; (III) results of nasal swabs for RDT assay. The two samples (nasopharyngeal and nasal swab) were obtained sequentially from patients with suspected COVID-19 disease who presented at least one sign or symptom, and from asymptomatic patients (no signs/symptoms related to COVID-19), but who required hospitalization for other issues.

### 2.2. SARS-CoV-2 RNA Detection

Nasopharyngeal swabs were collected in Universal Transport Medium for viruses, chlamydia, mycoplasma, and ureaplasma (Copan, Brescia, Italy). SARS-CoV-2 detection was performed with the Alinity m SARS-CoV-2 AMP Kit according to the manufacturer’s instructions [18] on the Abbott Alinity m system (Abbott Molecular Inc., Des Plaines, IL, USA). The RT-qPCR assay herein reported is a dual-target test based on the RdRp and N genes, with a limit of detection (LoD) of 100 copies/mL of SARS-CoV-2. The details of the procedure have been described in a previous work [19].

### 2.3. SARS-CoV-2 Antigen Detection

Detection of SARS-CoV-2 nucleocapsid protein antigen from nasal swabs was performed using the LumiraDx SARS-CoV-2 Ag Test [20], a rapid immunofluorescence assay based on microfluidic technology on the LumiraDx Platform (LumiraDx Limited, Stirling, UK). Briefly, a sample was collected with a standard dry swab of both nostrils of the patient and eluted into an extraction buffer vial. Then, a drop of sample was added to the test strip containing the dried reagents. After 12 min, the test result (positive or negative) was reported by instrumental fluorescence measurement.

### 2.4. Statistical Analysis

Categorical variables were described as absolute numbers and percentages, while continuous variables were reported as median and interquartile range (IQR). Sensitivity, specificity, positive predictive value (PPV), negative predictive value (NPV), positive likelihood ratio (LR+), negative likelihood ratio (LR−) with 95% confidence intervals (95% CI), and agreement of Cohen’s kappa between the antigen test and RT-qPCR were calculated according to Hazra and Gogtay [21]. The chi-square test was used to compare the distribution of categorical variables. Median cycle threshold (Ct) values were calculated and compared with the Mann–Whitney U test. The significance level was assessed at *p* ≤ 0.05. Statistical analysis was performed with SPSS software, version 17.0 (SPSS Inc., Armonk, NY, USA).

## 3. Results

A total of 375 patients were included in this study. The median age was 5 years (IQR: 1–11) and the male/female ratio was 1.4:1 (males: 219/375, 58.4%; females: 156/375, 41.6%); 36% (135/375) of patients reported signs or symptoms related to COVID-19 disease (Table 1). Among the 27 RT-qPCR-positive patients, 17/27 (63%) were asymptomatic. Furthermore, the median age of this subset was similar to that of those in the overall analysis (5 years, IQR: 0–10) and no significant difference was found between symptomatic/asymptomatic subsets and RT-qPCR results (positive symptomatic RT-qPCR: 10/135–7.4%; positive asymptomatic RT-qPCR: 17/240–7.1%; chi-square: 0.01; *p* = 0.920).

### 3.1. Overall Diagnostic Accuracy

The comparison between the two tests in both the symptomatic and asymptomatic subsets (overall population) is described in Table 2. Overall, 27/375 (7.2%) antigens tested positive as did 27/375 (7.2%) molecular tests, although not for the same patients.

Overall, we observed an estimated disease prevalence of 7.2% and the following results: 66.6% (95% CI: 46.0–82.7) sensitivity; 97.4% (95% CI: 94.9–98.7) specificity; 66.6% (95% CI: 46.0–82.7) PPV; 97.4% (95% CI: 94.9–98.7) NPV; 25.8 (95% CI: 12.8–51.8) LR+; 0.34 (95% CI: 0.20–0.58) LR−; Cohen’s kappa: 0.641 (95% CI: 0.487–0.795; *p* < 0.0001).

Focusing on the 27/375 swabs positive with the molecular assay and comparing the median Ct value between the RDT-positive subset (18/27, 66.7%) and the RDT-negative one (9/27, 33.3%), a significant difference was observed (RDT-positive subset: 17.7, IQR: 13.9–19.2; RDT-negative subset: 24.6, IQR: 18.4–33.9; *p* = 0.012). The differences between the abovementioned groups are reported in Figure 1.

### 3.2. Diagnostic Accuracy in Symptomatic Patients

Comparing the two tests in the subgroup of patients reporting at least one sign/symptom compatible with COVID-19 disease (n = 135, Table 2), we observed an estimated prevalence of disease of 7.4% (10/135) and the following results: 80% (95% CI: 44.2–96.5) sensitivity; 95.2% (95% CI: 89.4–98.0) specificity; 57.1% (95% CI: 29.6–81.2) PPV; 98.3% (95% CI: 93.5–99.7) NPV; 16.6 (95% CI: 7.19–38.6) LR+; 0.21 (95% CI: 0.06–0.73) LR−. Cohen’s kappa: 0.635 (95% CI: 0.403–0.867; *p* = 0.001). Evaluating the Ct of the 10/135 positive swabs by molecular assay and comparing the median Ct value between the RDT-positive samples (8/10, 80%) and the RDT-negative ones (2/10, 20%), no significant difference was found (*p* = 0.602). The median Ct values for RDT-positive and RDT-negative specimens were 14.9 (IQR: 13.1–17.5) and 18.4 (IQR: 13.8–23.1), respectively (Figure 2).

### 3.3. Diagnostic Accuracy in Asymptomatic Patients

The analysis performed in the subgroup of patients without any sign/symptom related to COVID-19 disease at admission (n = 240, Table 2) resulted in an estimated prevalence of 7.1% and in the following data: 58.8% (95% CI: 33.5–80.6) sensitivity; 98.7% (95% CI: 95.8–99.7) specificity; 76.9% (95% CI: 45.9–93.8) PPV; 96.9% (95% CI: 93.5–98.6) NPV; 43.7 (95% CI: 13.3–144.0) LR+; 0.42 (95% CI: 0.24–0.74) LR−; Cohen’s kappa: 0.645 (95% CI: 0.441–0.849; *p* < 0.0001).

Considering the 17/240 (7.1%) positive samples with the Rt-qPCR assay and the related median Ct values in the RDT-positive subset (10/17, 58.8%) and RDT-negative one (7/17, 41.2%), a significant difference was observed (RDT-positive subset: 18.9, IQR: 17.2–21.2; RDT-negative subset: 33.3, IQR: 21.5–36.5; *p* = 0.032) (Figure 3).

## 4. Discussion

Our data showed a high specificity (97.4%) of the RDT assay in a pediatric population with an estimated prevalence of 7.2%. The probability of a positive RDT result when RT-qPCR tested positive was almost 26 times higher than in the RDT-positive/RT-qPCR-negative condition. Otherwise, focusing on the asymptomatic subset, the agreement between positive RDT and RT-qPCR results increased almost 44-fold, while for the symptomatic subgroup it was lower (near 17-fold) (Table 2).

We observed an overall false positive rate of 2.4% for the RDT assay, with an increase in symptomatic patients (4.5%) (Table 2). Despite its user-friendliness, it should be noted that the RDT available as POCT must be performed according to the manufacturer’s instructions in order to obtain accurate test results [22]. Moreover, a potential cross-reaction between HKU-1 and SARS-CoV-2 was possible due to the low homology between the nucleocapsid proteins of the two coronaviruses [23]. These features could be responsible for the false-positive rate observed in our study, especially in the symptomatic subset. The relatively lower PPV in the considered patient groups (Table 2) was potentially related to the prevalence of the disease observed in the population of our work, where even a small number of false-positive RDT results out of a total of equally low number of RT-qPCR-positive patients could lead to a PPV drop [24].

The sensitivity values reported in the present study could be explained through the comparative analysis of Ct in RDT-positive/-negative specimens that tested positive with the RT-qPCR assay (n = 27). Indeed, the median Ct value of negative RDT samples was significantly higher than that of positive ones both in the overall analysis and in the subset including asymptomatic patients (Figure 1 and Figure 3). Nevertheless, the sensitivity values herein observed (overall: 66.6%; symptomatic subset: 80%; asymptomatic subset: 58.8%) were higher than the 30.2% reported by Scohy et al. [25] and the 50% described by Lambert-Niclot et al. [26]. Focusing on the pediatric population, González-Donapetry et al. [12] evaluated a POCT for diagnosis of SARS-CoV-2 infection compared to RT-qPCR in symptomatic patients and found an overall sensitivity of 77.7%, lower than the results of our study (Table 2). Another analysis, performed on 199 symptomatic children during a period when prevalence was high (26%) [13], reported a sensitivity of 85%, similar to that observed here (80%). Otherwise, in another study [14] carried out on 1620 symptomatic pediatric patients in a low-prevalence setting (5%), an overall sensitivity of 45.4% was observed. Overall, the RDT showed moderate sensitivity (65.9%, 95% CI: 52.8–77.0%) and high specificity (99.9%) for the detection of SARS-CoV-2 in children [27]. Nevertheless, it should be noted that all the abovementioned studies were based on comparison of first- and second-generation immunochromatographic tests to the RT-qPCR assay. Fewer data were available for third-generation microfluidic tests: an evaluation performed in asymptomatic adults and children demonstrated a PPA of 82.1% (95% CI: 64.4–92.1%) and an NPV of 100% compared to the molecular method [28]. Similar results were observed in this work for the asymptomatic subset (PPA: 76.9%, 95% CI: 45.9–93.8; NPV: 96.9%, 95% CI: 93.5–98.6). For the RT-qPCR-positive samples tested in the present work, no variant analysis was available; however, a survey carried out by the Ministry of Health on 24 August 2021 reported an absolute prevalence (100%) of the delta variant, lineage B.1.617.2 throughout the Piedmont territory [29].

Data shown in the present work were collected from a single-center evaluation and the results may not be fully applicable to other settings. The main limitations were the inability to evaluate the viability of the SARS-CoV-2 virus in negative RDT swabs and the bias resulting from the different sample collection procedures. Finally, no details were available regarding the days following clinical onset for symptomatic patients with COVID-19; therefore, it was not possible to evaluate the impact of this factor.

## 5. Conclusions

Compared to RT-qPCR, the third-generation microfluidic-based antigen test showed higher specificity (>95%) in the pediatric population, thus representing a suitable POCT in a clinical setting with low prevalence of COVID-19.

## Figures and Tables

**Figure 1 viruses-16-00041-f001:**
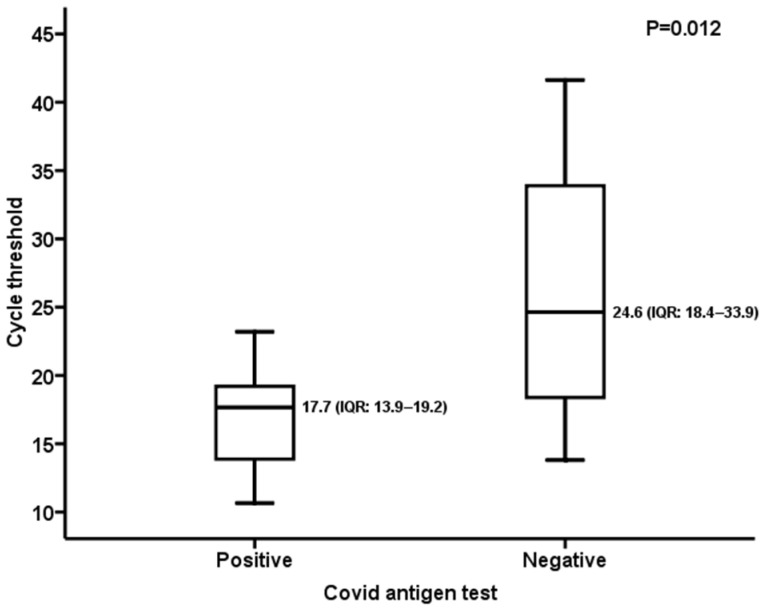
Overall population: RT-qPCR cycle threshold value in RDT-positive/-negative samples.

**Figure 2 viruses-16-00041-f002:**
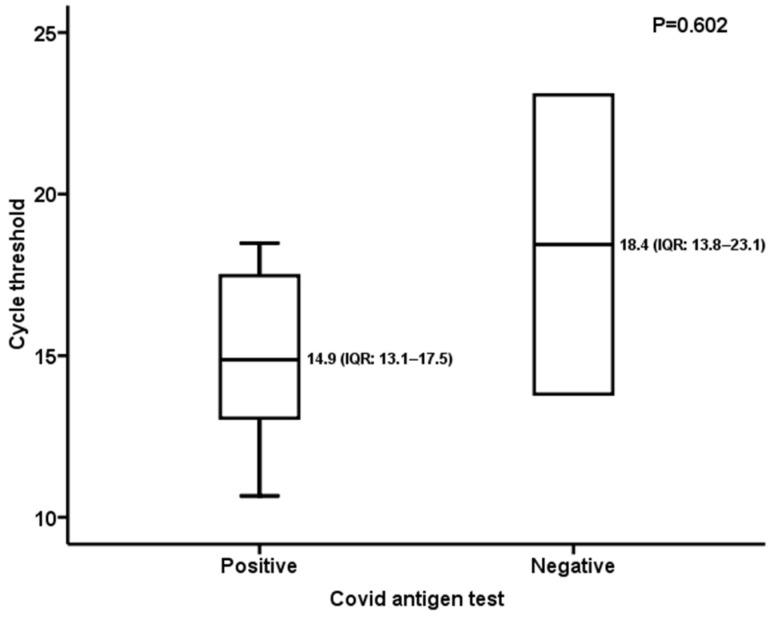
Symptomatic patients: RT-qPCR cycle threshold value in RDT-positive/-negative samples.

**Figure 3 viruses-16-00041-f003:**
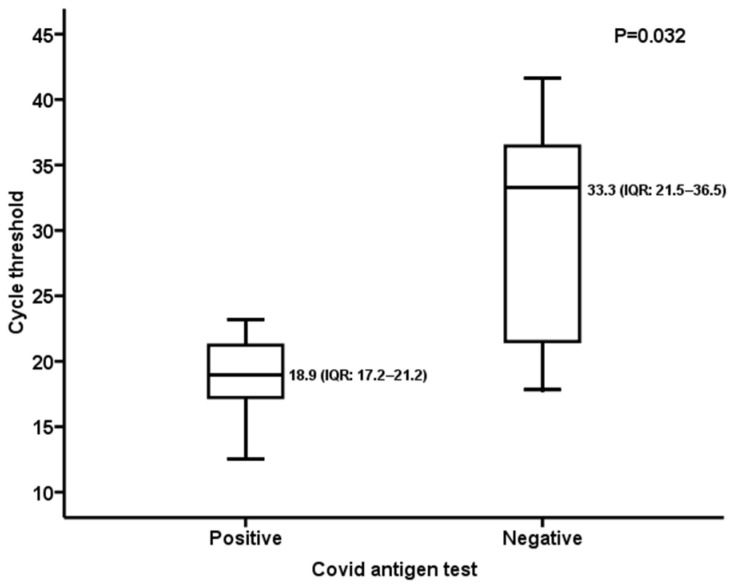
Asymptomatic patients: RT-qPCR cycle threshold value in RDT-positive/negative samples.

**Table 1 viruses-16-00041-t001:** Clinical feature of population (n = 375) according to RT-qPCR results.

Sign/Symptom	N (%)	RT-qPCR	
		Positive (%)	Negative (%)
Asymptomatic	240 (64)	17 (7.1)	223 (92.9)
Symptomatic	135 (36)	10 (7.4)	125 (92.6)
Body temperature ≥ 37.5 °C	77 (20.5)	6 (7.8)	71 (92.2)
Cough	16 (4.3)	1 (6.2)	15 (93.8)
Dyspnea	10 (2.6)	0 (0)	10 (100)
Headache	8 (2.1)	2 (25)	6 (75)
Pharyngodynia	5 (1.3)	1 (20)	4 (80)
Asthenia	5 (1.3)	0 (0)	5 (100)
Rhinorrhea	1 (0.3)	0 (0)	1 (100)
Chest pain	2 (0.6)	0 (0)	2 (100)
Abdominal pain	23 (6.1)	0 (0)	23 (100)
Nausea	5 (1.3)	0 (0)	5 (100)
Diarrhea	7 (1.9)	0 (0)	7 (100)

**Table 2 viruses-16-00041-t002:** Comparison of diagnostic accuracy of rapid antigen test with RT-qPCR.

		RT-qPCR		
		Positive (%)	Negative (%)	Total (%)
RDT	Overall population (n = 375)			
	Positive (%)	18 (4.8)	9 (2.4)	27 (7.2)
	Negative (%)	9 (2.4)	339 (90.4)	348 (92.8)
	Total (%)	27 (7.2)	348 (92.8)	375 (100)
	Symptomatic (n = 135)			
	Positive (%)	8 (5.9)	6 (4.5)	14 (10.4)
	Negative (%)	2 (1.5)	119 (88.1)	121 (89.6)
	Total (%)	10 (7.4)	125 (92.6)	135 (100)
	Asymptomatic (n = 240)			
	Positive (%)	10 (4.1)	3 (1.3)	13 (5.4)
	Negative (%)	7 (2.9)	220 (91.7)	227 (94.6)
	Total (%)	17 (7.1)	223 (92.9)	240 (100)
	Rapid antigen test performance			
		%	95% CI	
	Overall population (n = 375)			
	Sensitivity	66.6	46.0–82.7	
	Specificity	97.4	94.9–98.7	
	PPV	66.6	46.0–82.7	
	NPV	97.4	94.9–98.7	
	LR+	25.8	12.8–51.8	
	LR−	0.34	0.20–0.58	
	Symptomatic (n = 135)			
	Sensitivity	80.0	44.2–96.5	
	Specificity	95.2	89.4–98.0	
	PPV	57.1	29.6–81.2	
	NPV	98.3	93.5–99.7	
	LR+	16.6	7.19–38.6	
	LR−	0.21	0.06–0.73	
	Asymptomatic (n = 240)			
	Sensitivity	58.8	33.5–80.6	
	Specificity	98.7	95.8–99.7	
	PPV	76.9	45.9–93.8	
	NPV	96.9	93.5–98.6	
	LR+	43.7	13.3–144.0	
	LR−	0.42	0.24–0.74	

Data are reported as absolute numbers and percentage (%); RT-qPCR: real-time reverse transcriptase polymerase chain reaction; CI: confidence interval; PPV: positive predictive value; NPV: negative predictive value; LR+: positive likelihood ratio; LR−: negative likelihood ratio.

## Data Availability

The data contained in this manuscript are available upon request.

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
