# Peer review of "Application of a SARS-CoV-2 Antigen Rapid Immunoassay Based on Active Microfluidic Technology in a Setting of Children and Young Adults"

_viruses, 2023, doi:10.3390/v16010041_

Round 1
Reviewer 1 Report
Comments and Suggestions for Authors
The article exhibited an application of a SARS-CoV-2 antigen rapid immunoassay based on active microfluidic technology. The article can be considered for publication in " Viruses " after revising the following questions. The comments are below.
1) In the introduction, the author mainly describes the hazards of SARS-CoV-2 and the current detection methods. In fact, as can be seen from the title of the article, microfluidic technology is also worth emphasizing. We suggest that the author explain the application examples of microfluidic in the detection of SARS-CoV-2 in the introduction and compare them with the methods in the paper.
Micromachines http://dx.doi.org/10.3390/mi13081238
2) More figures about microfluidic technology and detection principles should be drawn to help readers understand the innovative results made by the author in this study
3) Offer a bit more context or detail regarding the symptoms considered possibly associated with COVID-19. This could include specifying the range of symptoms assessed or referring to the criteria used to categorize patients into symptomatic and asymptomatic groups.
4) Discuss the relatively low rate of false-positive results in RT-qPCR negative patients and its significance in terms of the test's reliability, especially in a low-prevalence setting.
5) Provide a comparative analysis with other studies assessing similar diagnostic methods in pediatric populations. Explain differences in sensitivity values observed in previous studies and how they relate to the findings of the current research.
Author Response
We are very grateful for the advices and the comments provided by the reviewers in order to improve our manuscript. We have taken in account all these corrections in the revised version of the manuscript. Moreover, the text has been extensively corrected to improve its readability and comprehension. Below I have stated our response to each of the suggestions made by the reviewer. We hope that the modifications made in the revised manuscript and that the way we addressed the comments made by you and the reviewers meet your expectations. Otherwise, we are open to new improvements.
1. In the introduction, the author mainly describes the hazards of SARS-CoV-2 and the current detection methods. In fact, as can be seen from the title of the article, microfluidic technology is also worth emphasizing. We suggest that the author explain the application examples of microfluidic in the detection of SARS-CoV-2 in the introduction and compare them with the methods in the paper. (Micromachines http://dx.doi.org/10.3390/mi13081238)
We thank the reviewer for this interesting observation. We added an updated short paragraph about the different technologies used as RDTs (from first to fourth generation), with a focus on microfluidic assays and their advantages (lines 45-57).
2. More figures about microfluidic technology and detection principles should be drawn to help readers understand the innovative results made by the author in this study
As abovementioned, according to reviewer suggestion, we added a dedicated paragraph reporting the pivotal features of a microfluidic RDT. No figures have been added since, in our opinion, they were more suitable for a review or meta-analysis.
3. Offer a bit more context or detail regarding the symptoms considered possibly associated with COVID-19. This could include specifying the range of symptoms assessed or referring to the criteria used to categorize patients into symptomatic and asymptomatic groups.
Each patient presenting at least one sign/symptom related to COVID-19 (table 1) was considered as potentially affected and included in the symptomatic subset. Data collected and criteria considered for classification of patients enrolled in the present study were better mentioned in material and methods (lines 69-78).
4. Discuss the relatively low rate of false-positive results in RT-qPCR negative patients and its significance in terms of the test's reliability, especially in a low-prevalence setting.
We agree with the reviewer for this observation. A paragraph in the discussion section (lines 204-210) was added to deepen the possible reasons responsible for observed RDT false positive results. We think that the cross-reactivity between HKU-1 and SARS-CoV-2 must be better investigated in the future.
5. Provide a comparative analysis with other studies assessing similar diagnostic methods in pediatric populations. Explain differences in sensitivity values observed in previous studies and how they relate to the findings of the current research.
We thank the reviewer for the suggestion and revised the discussion with other two references focused on comparison between RDT assay and RT-qPCR (lines 222-237). Besides yet reported studies, an updated meta-analysis was mentioned in order to better correlate our data with other studies.
Reviewer 2 Report
Comments and Suggestions for Authors
The manuscript submitted to Viruses demonstrated successful application of antigen rapid immunoassay test based on microfluidic technology for SAR-CoV-2 in pediatric population and low-prevalence test. The manuscript is well written with clear presentation of data. Some major changes are needed to improve this manuscript.
1. There are multiple sensitive antigen-based test available. It would be good to add in the introduction some of the other highly sensitive antigen-based kits tested for SARS-CoV-2 with references.
2. The comparison of this novel method based on microfluidic system is done with RT-qPCR technique only. It will be good to compare this kit across other highly sensitive kits commercially available for detection of SARS-CoV-2. It is expected to add any such results in the manuscript which shows sensitive detection by this microfluidic based kit over the other 2-3 commercially available kits.
Author Response
We are very grateful for the advices and the comments provided by the reviewers in order to improve our manuscript. We have taken in account all these corrections in the revised version of the manuscript. Moreover, the text has been extensively corrected to improve its readability and comprehension. Below I have stated our response to each of the suggestions made by the reviewer. We hope that the modifications made in the revised manuscript and that the way we addressed the comments made by you and the reviewers meet your expectations. Otherwise, we are open to new improvements.
1. There are multiple sensitive antigen-based test available. It would be good to add in the introduction some of the other highly sensitive antigen-based kits tested for SARS-CoV-2 with references.
We thank the reviewer for this interesting observation. A short paragraph about RDT (from first to fourth generation) and overall data about their sensitivity and specificity was added in the introduction. Moreover, according also to suggestion of other reviewer, we reported a brief focus about microfluidic RDT with their features and advantages (45-57).
2. The comparison of this novel method based on microfluidic system is done with RT-qPCR technique only. It will be good to compare this kit across other highly sensitive kits commercially available for detection of SARS-CoV-2. It is expected to add any such results in the manuscript which shows sensitive detection by this microfluidic based kit over the other 2-3 commercially available kits.
The idea of comparison with other kits is very interesting, although no other tests were available in our setting and a correlation between different RDT was not affordable. However, we updated the manuscript with other references to provide a better correlation of our data with the other studies (lines 229-237). We may think to do a future review or metanalysis about this purpose. An example of in-depth analysis of antigenic test available with their performances can be found to this link: https://www.eurosurveillance.org/content/10.2807/1560-7917.ES.2021.26.44.2100441
Reviewer 3 Report
Comments and Suggestions for Authors
1. Regarding the sample collection, were two nasal swabs used simultaneously or subsequently to collect the nasopharyngeal cells? Especially for the latter, how do we determine which sample was assigned for RT-PCR and RDT? Please clarify the swab sample collection procedure.
2. The authors described that studies focused on evaluating RDTs' diagnostic performance in pediatric populations are scarce; however, at least two relevant published studies are not cited in the reference list.
3. The description of the whole population is not appropriate. It is easy to confuse the general population.
4. We had a prevalence of disease of 7.2%; the “estimated” prevalence will be suitable.
5. Why did the authors not report the sensitivity in the Abstract?
6. Although this new active microfluidic-based antigen test showed a near 100% specificity in a pediatric population, this study just compared with the gold standard; they did not compare their results with other existing RDT methods.
7. Information regarding days after clinical onset for symptomatic COVID-19 patients was not available. Why is this so? Especially for those symptomatic patients.
Comments on the Quality of English LanguageEnglish editing is recommended.
Author Response
We are very grateful for the advices and the comments provided by the reviewers in order to improve our manuscript. We have taken in account all these corrections in the revised version of the manuscript. Moreover, the text has been extensively corrected to improve its readability and comprehension. Below I have stated our response to each of the suggestions made by the reviewer. We hope that the modifications made in the revised manuscript and that the way we addressed the comments made by you and the reviewers meet your expectations. Otherwise, we are open to new improvements.
1. Regarding the sample collection, were two nasal swabs used simultaneously or subsequently to collect the nasopharyngeal cells? Especially for the latter, how do we determine which sample was assigned for RT-PCR and RDT? Please clarify the swab sample collection procedure.
Samples were collected at the same time as following: first, swab from nasopharyngeal, then the nasal swab from patient nostrils. For each sample type a dedicated swab of suitable diameter was used. Criteria of sampling were better specified at lines 69-78.
2. The authors described that studies focused on evaluating RDTs' diagnostic performance in pediatric populations are scarce; however, at least two relevant published studies are not cited in the reference list.
We agree with the reviewer suggestion and reformulated the sentences since other studies about RDT in pediatric population were added from our first analysis. A further reference was added in the introduction in order to suggest the role of children in COVID-19 pandemic and why studies about this cohort were fewer compared to adult population (lines 58-61). Moreover, an updated meta-analysis was mentioned in the discussion, besides yet reported studies, to better correlate our data with other studies (lines 229-237).
3. The description of the whole population is not appropriate. It is easy to confuse the general population.
Corrected along the manuscript as “overall population”.
4. We had a prevalence of disease of 7.2%; the “estimated” prevalence will be suitable.
Corrected along the whole manuscript as suggested.
5. Why did the authors not report the sensitivity in the Abstract?
We apologize for the mistake and added sensitivity values in the abstract in order to provide a complete overview of RDT performances. However, the most important data were related to specificity since overall sensitivity and related to the two reported subgroups (symptomatic/asymptomatic) was not higher.
6. Although this new active microfluidic-based antigen test showed a near 100% specificity in a pediatric population, this study just compared with the gold standard; they did not compare their results with other existing RDT methods.
Since no other tests were available in our setting, an in-house correlation was not affordable. However, we added other references, one of which focused on microfluidic RDT in a mixed population of adult and children (lines 232-237).
7. Information regarding days after clinical onset for symptomatic COVID-19 patients was not available. Why is this so? Especially for those symptomatic patients.
We agree with the reviewer observation. However, data herein reported were related to symptom observed at the admission in ED and for most of them the onset was not available or not collected. Otherwise, we observed some asymptomatic patients with a previously reported history of possible signs or symptoms. However, they were not observed by clinicians at the admission in ED and for this reason classified into the asymptomatic subset. For these reasons and to avoid potential bias, the two subsets were obtained only on the basis of signs/symptoms observed when admitted to the ED. This limitation was mentioned at the end of discussion.
Round 2
Reviewer 2 Report
Comments and Suggestions for Authors
The Author's has included existing RDT and their sensitivity with relevant references in the updated version. There is also addition of statement that highlights better correlation of their data with other studies in discussion. Overall readability of the manuscript is improved. I would highly recommend to accept this manuscript for publication.